# Identification of Genetic Loci for Rice Seedling Mesocotyl Elongation in Both Natural and Artificial Segregating Populations

**DOI:** 10.3390/plants12142743

**Published:** 2023-07-24

**Authors:** Fangjun Feng, Xiaosong Ma, Ming Yan, Hong Zhang, Daoliang Mei, Peiqing Fan, Xiaoyan Xu, Chunlong Wei, Qiaojun Lou, Tianfei Li, Hongyan Liu, Lijun Luo, Hanwei Mei

**Affiliations:** 1Shanghai Agrobiological Gene Center, Shanghai 201106, China; ffj@sagc.org.cn (F.F.); mxs09@sagc.org.cn (X.M.);; 2Key Laboratory of Grain Crop Genetic Resources Evaluation and Utilization, Ministry of Agriculture and Rural Affairs, Shanghai 201106, China; 3National Key Laboratory of Crop Genetic Improvement, Huazhong Agricultural University, Wuhan 430070, China; 4Anji Administrative Station of Water and Soil Conservation, Huzhou 313300, China

**Keywords:** *Oryza sativa* L., mesocotyl elongation, GWAS, linkage mapping, dry direct sowing rice (DDSR)

## Abstract

Mesocotyl elongation of rice seedlings is a key trait for deep sowing tolerance and well seedling establishment in dry direct sowing rice (DDSR) production. Subsets of the Rice Diversity Panel 1 (RDP1, 294 accessions) and Hanyou 73 (HY73) recombinant inbred line (RIL) population (312 lines) were screened for mesocotyl length (ML) via dark germination. Six RDP1 accessions (Phudugey, Kasalath, CA902B21, Surjamkuhi, Djimoron, and Goria) had an ML longer than 10 cm, with the other 19 accessions being over 4 cm. A GWAS in RDP1 detected 118 associated SNPs on all 12 chromosomes using a threshold of FDR-adjusted *p* < 0.05, including 11 SNPs on chromosomes 1, 4, 5, 7, 10, and 12 declared by −log_10_(*P*) > 5.868 as the Bonferroni-corrected threshold. Using phenotypic data of three successive trials and a high-density bin map from resequencing genotypic data, four to six QTLs were detected on chromosomes 1, 2, 5, 6, and 10, including three loci repeatedly mapped for ML from two or three replicated trials. Candidate genes were predicted from the chromosomal regions covered by the associated LD blocks and the confidence intervals (CIs) of QTLs and partially validated by the dynamic RNA-seq data in the mesocotyl along different periods of light exposure. Potential strategies of donor parent selection for seedling establishment in DDSR breeding were discussed.

## 1. Introduction

Rice production is facing critical challenges due to shortages of irrigation water and labor resources; rapid increases in resource and labor costs and relatively stable grain prices have led to poor profits and deficits in the industry. Seedling transplanting in puddled fields is still the predominant cultivation system in major rice production countries, especially in high-yielding acreages. This traditional system, more precisely termed as transplanted–flooded rice by Liu et al. (2015), is water- and labor-consuming [1]. Field puddling and seedling transplanting consume up to 30% of the total water for the whole cropping season [2], which is not biologically necessary for the growth and development of rice plants. In rainfed rice cultivation, field preparation and transplanting may fail or be severely delayed when rainfall is inadequate or postponed in the window of crop rotation [3,4]. Although mechanized transplanting and wet direct sowing have been widely used in rice production, dry direct sowing rice (DDSR), especially when mechanized, has been suggested as an optimal rice cultivation system to save natural and labor resources and to consequently increase farmers’ incomes [1,5]. Based on a positive review of previous reports and their own experimental results, Liu et al. (2015) nominated DDSR as a sustainable and feasible alternative to transplanted rice as it can produce a similar (or even higher) grain yield, largely save water and labor resources, and enhance water and nitrogen use efficiency [1].

The adaptation of rice varieties to dry direct sowing and the proper management of soil conditions, sowing time, and depth are key issues to assure rapid and full seedling establishment, which is beneficial to weed competition, rice plant growth and development, and the final grain yield. Rice varieties with less sensitivity to sowing depth, i.e., tolerance to deep sowing, are beneficial for allowing the seeds to germinate rapidly using the higher moisture in deep soil layers, for which mesocotyl elongation is a key underlying characteristic [6,7]. Screening several sets of rice germplasm collections has not only identified accessions with a largely elongated mesocotyl, but also showed the low proportion of such accessions in those populations. It was also found that there was a relatively higher percentage of accessions with elongated mesocotyls in landraces than in improved semidwarf varieties and in upland rice than in paddy rice [8,9,10,11,12].

Linkage analysis and genome-wide association studies (GWASs) have been widely adopted to identify loci significantly associated with important and complex agronomic traits in rice. Dozens of QTLs have been identified for mesocotyl elongation in different populations. Five QTLs have been mapped for mesocotyl elongation on chromosomes 1, 3, 5, and 7 using an F_3_ population developed from a cross between a *japonica* cultivar, Labelle, and an *indica* cultivar, Black Gora [9]. Thirteen loci for mesocotyl elongation on chromosomes 1, 3, 4, 5, 6, and 9 were identified using a GWAS in our previous study [12]. Eight QTLs on chromosomes 1, 3, 6, 7, 8, and 12 were identified for mesocotyl elongation using a doubled-haploid population from a cross between IR64 and Azucena [13]. Eleven QTLs were identified for mesocotyl elongation on chromosomes 1, 3, 4, 5, 6, 9, and 11 using an RIL population [14]. Lee et al. (2012) detected five QTLs for mesocotyl elongation on chromosomes 1, 3, 7, 9, and 12 [15]. Liu et al. (2020) reported 16 unique loci for mesocotyl elongation through a GWAS [16]. Wang et al. (2021) identified 14 QTLs for mesocotyl elongation via QTL sequencing using 12 F_2_ populations, and five unique QTLs with a GWAS using two diverse panels [17]. Gothe et al. (2022) also identified eight QTLs for mesocotyl elongation using a GWAS [18]. Several genes were characterized that regulated mesocotyl elongation via different molecular mechanisms, including examples like *GY1* [19], *OsGSK2* [20], *OsPAO5* [21], and *OsSMAX1* [22].

A large collection of global diverse rice germplasm accessions and its genotypic dataset based on high-density SNP markers, such as the open-access resources of the Rice Diversity Panels (RDP1 and RDP2) and genotypic data generated with a high-density rice array, provide great opportunities for the international rice research community to investigate the genetic basis of various phenotypic traits [23,24,25,26,27,28]. The screening of mesocotyl elongation in a subset of RDP1 showed rich genetic variation, supporting GWAS for this characteristic.

Seed vigor is usually defined as the ability of genotypes or seed samples to obtain strong seedlings via quick germination and fast growth at the early stage. Drought-resistant rice cultivars have been shown to have higher seed vigor and better seedling establishment after dry direct sowing than popular irrigated rice cultivars, especially under stress conditions like drought and phosphorus deficiency [29]. Early studies found that mesocotyl elongation is the key characteristic contributing to seedling vigor under aerobic soil conditions, even though elongations of both the mesocotyl and the coleoptile are anatomically beneficial to seedling emergence [7,30]. On the other hand, the seedling vigor of a hybrid (F_1_) is usually higher than the average performance of two parental lines, even higher than the better parent, showing heterosis at the early seedling stage. Among reciprocal hybrids and their parental lines (three *japonica* cultivars and the *indica* cultivar IR50), significant correlation was observed between the emergence index of the second leaf (LEI) and germination index (GI): LEI = 1.561 + 0.793 × GI (*r* = 0.890 ***). There was high-level midparent heterosis (MPH) for characteristics related to seedling growth in most cases of hybrid combinations, especially for subspecific hybrids. For example, the leaf area index had an MPH of 2.4–27.6% among those hybrids at 23 days after sowing (DAS) [30]. As a commercial hybrid variety of water-saving and drought-resistant rice (WDR) [31,32], Hanyou 73 (HY73, Huhan7A × Hanhui 3) has been widely cultivated in drought-prone areas in China, showing promising potential to save labor and water resources in DDSR production. Superior seedling emergence and moderate mesocotyl elongation of HY73 were also observed in the field trial of DDSR with a sowing depth as high as 8–10 cm. An interesting question is whether the genetic basis of mesocotyl elongation or the heterosis of the hybrid determines the superior seedling emergence under dry sowing conditions. A set of recombinant inbred lines were developed from the hybrid of HY73. Linkage mapping of mesocotyl elongation was conducted using a high-density bin map from genotyping via the resequencing approach.

In this study, the associated loci and QTL in two different populations, together with candidate genes, are jointly analyzed to identify important genomic regions for mesocotyl elongation. Based on the seedling morphological and agronomical features of the germplasm accessions with the highest mesocotyl lengths, strategies for selecting the favored donor parental lines in DDSR breeding are also discussed.

## 2. Results

### 2.1. Phenotypic Variation in Mesocotyl Elongation in RDP1 and HY73 RILs

In the subset of RDP1, a wide range of mesocotyl lengths were observed, while a large proportion of the accessions had low mesocotyl elongation ability in darkness, showing a very high peak of frequency at the left end (ML < 1 cm) together with a long tail at the right end (Figure 1A). The ANOVA results showed highly significant variation in mesocotyl elongation among the accessions (Table 1). Six accessions from five different countries had MLs higher than 10cm, including Phudugey (19.0 cm, Bhutan), Kasalath (14.1 cm, India), CA902B21 (13.6 cm, Chad), Surjamkuhi (12.3 cm, India), Djimoron (12.3 cm, Guinea), and Goria (10.6 cm, Bangladesh). There were another 19 accessions with MLs longer than 4 cm, including Karkati 87 (Bangladesh), MTU 9 (India), PTB 30 (India), Arias (Indonesia), Azerbaidjanica (Azerbaijan), P373 (Pakistan), Jhona 349 (India), N12 (India), NSF-TV13 (unknown), Sinampaga selection (Philippines), Basmati 217 (India), WC 6 (China), Basmati (Pakistan), T26 (India), Rondo (China), DM56 (Bangladesh), Lemont (USA), Sadri Belyi (Azerbaijan), and 9524 (India).

For the RIL population from the hybrid HY73, the seedlings of two parental lines, Huhan7A and Hanhui 3, had mesocotyl lengths of about 2.5 cm and 4.0 cm, respectively. The subset of the RIL population had nearly normal distribution and ranges of around 1 cm to 9 cm, suggesting noticeable transgressive segregation in mesocotyl lengths, especially on the side of high phenotypic values (Figure 1B). The ANOVA results showed highly significant variation among the lines (Table 1). There were also significant variations among the trials and line–trial interactions, suggesting a high sensitivity in mesocotyl elongation phenotyping to environments.

As measured in this experiment and another sand culture trial in 2019 with a sowing depth of 8 cm, the seedlings of the hybrid (HY73, F_1_) had an average elongated mesocotyl length of about 2.5 cm, while the sterile maternal parent (Huhan7A) and paternal restorer line (Hanhui 3) had mesocotyl lengths of 2–2.5 cm and 3.5–4 cm, respectively (Figure 2). This result supports the recessive (or at least semirecessive) inheritance of mesocotyl elongation in rice.

### 2.2. Genome-Wide Association Study in RDP1

A total of 118 associated SNPs were declared on all 12 rice chromosomes using the threshold of FDR-adjusted *p* < 0.05. Among them, 11 SNPs reached the stringent threshold of −log_10_ (0.05/36901) = 5.868, according to the Bonferroni correction for multiple tests (Appendix A). There were at least 17 peaks of −log_10_ (*p*) values showing association signals along the rice genome, including three peaks on each of chromosomes 1 and 7; two on each of chromosomes 6, 11, and 12; and one on each of chromosomes 2–5 and 10. Eleven SNP markers with high −log_10_ (*p*) values fell into the six peaks of association signals on rice chromosomes 1, 4, 5, 7, 10, and 12 (Figure 3).

### 2.3. Linkage Mapping in HY73 RILs

Using the phenotypic values measured in three successive trials (ML1, ML2, and ML3), four, five, and six QTLs were detected for the elongated mesocotyl length, respectively (Table 2), including three on chromosome 1 and one on chromosome 9 for ML1; three on chromosome 1 and one on each of chromosomes 5 and 6 for ML2; and three on chromosome 1 and one on each of chromosomes 2, 6, and 10 for ML3. Among them, two identical intervals, Chr1_bin348–Chr1_bin350 and Chr6_bin6431–Chr6_bin6433, were repeatedly mapped for ML1 and ML3 and ML2 and ML3, respectively. The superior allele of the QTL interval Chr1_bin348–Chr1_bin350 was derived from Hanhui 3, and the superior allele of the QTL interval Chr6_bin6431–Chr6_bin6433 was derived from Huhan 7A. The significant interval on chromosome 1 Chr1_bin147–Chr1_bin149 for ML2 was close to Chr1_bin140–Chr1_bin146 for ML3, and superior alleles were both derived from Hanhui 3. Three significant intervals, Chr1_bin1342–Chr1_bin1348 for ML1, Chr1_bin1322–Chr1_bin1327 for ML2, and Chr1_bin1396–Chr1_bin1398 for ML3, were located adjacent to one another on the end of chromosome 1 (Figure 4), of which the superior alleles were derived from Hanhui 3. The abovementioned colocated or adjacent QTLs had the same direction of additive effects (Table 2), suggesting one QTL for ML at each location.

### 2.4. Candidate Genes Predicted from Associated or Linked Genomic Regions

A primary candidate gene pool was developed by including 1954 annotated genes from either the LD blocks anchored by the associated SNPs in the RDP1 population or the CIs of QTLs detected in the HY73 RIL population (Appendix A). Among them, 64 candidate genes were validated from their dynamic expression patterns during mesocotyl growth blocking by light, i.e., DEGs in the mesocotyl tissues of the seedlings exposed to light for 20 min, 60 min, or 360 min vs. the seedlings in darkness (Appendix A) [33]. Three candidate genes were differentially expressed in all three light exposure periods, including *LOC_Os01g09100*, *LOC_Os07g03120*, and *LOC_Os09g37600*, annotated as OsWRKY10—a superfamily of TFs with WRKY and zinc finger domains, unknown expressed protein, and lysM domain-containing GPI-anchored protein precursor, respectively. Another 12 genes showed shifted expressions after light exposure for two time points, including 11 genes at 60 min light exposure together with the shorter (20 min) or longer (360 min) light exposure, and only one gene at both 20 min and 360 min light exposure (Appendix A). It is noticeable that the expression levels of all three genes were induced by light exposure for three time periods, while those DEGs detected at two time points also kept the same direction, i.e., upregulated or downregulated in both light treatments. 

## 3. Discussion

### 3.1. Genetic Variations in Mesocotyl Elongation Have Been Investigated in Different Rice Germplasm Populations

A total of 3677 rice accessions, or 3971 accessions if including the 294 accessions in this study, have been screened for mesocotyl elongation in 11 phenotypic experiments, as publicly reported since 1996 (Appendix A). These natural variation populations cover almost all taxonomic branches of the germplasm collections in *Oryza sativa* L., e.g., landraces or varieties, upland or lowland rice, traditional or modern semidwarf varieties, together with weedy rice and *O*. *nivara* accessions. Benefitting greatly from powerful technical solutions, several rice germplasm collections have been developed to enrich the genetic variation, i.e., to contain a high percentage of the total genetic variations in the whole gene pool in subsets with smaller population sizes, e.g., the mini-core collection of Chinese rice germplasm [34], RDP1 [23,24], and 3010 rice accessions in 3K-RGP [35]. Using the diverse rice germplasm collections presented in this study and previous reports [12,36], together with special collections like upland rice and weedy rice, we can be highly confident that phenotypic screening results up to now have covered the major proportion of the genetic variation for mesocotyl elongation in the gene pool of rice germplasm collections.

However, varied protocols have been employed in different phenotypic screening experiments, roughly following two main strategies. A few methods have been modified from the standard protocol of a seed germination test. Seedlings are cultured on moist filter papers, gel media, or within filter paper rolls in plastic boxes, bottles, or other containers with more space than Petri dishes; then, they are easily separated for ML measurement. Other methods try to cover the seeds with moist soil, sand, or humus layers, simulating direct sowing in the field. Absolute darkness should be maintained throughout the whole procedure as light exposure at the early stage of germination might inhibit mesocotyl growth. Additionally, as submergence or extremely high moisture promote coleoptile growth but inhibit mesocotyl elongation, soil or sand water content needs to be uniform and well controlled among the samples. The method of humus soil culture with a 6 cm sowing depth is used in ML phenotyping and has been suggested as an optimal protocol [16]. However, exposing seedling tips to light after emergence from a 6 cm depth will block the further growth of the mesocotyl, thus causing an underestimation of mesocotyl elongation ability above 4–5 cm, similar to the results measured from the 5 cm sand culture [12]. If the whole culture procedure is conducted in darkness, precisely controlling the covering depth seems unnecessary. To evaluate the mesocotyl elongation potential of germplasm accessions, dark germination should still be first choice, being the cheapest and quickest method. But soil or sand culture with a specific sowing depth, close to or slightly higher than the largest sowing depth required in rice production, would be a sufficient protocol for screening germplasm accessions or segregating populations in DDSR breeding programs. Allowing for sowing depth ranges that actually occur in the field, a sand culture with 8 cm depth was employed in the screening trials of seed bulks of inbred or BC progenies by the authors’ team and was quite effective in developing new lines with deep sowing tolerance. Using nylon net sheets to keep the seeds at a relatively equal depth and hold the seedlings after “digging out” at the end of the screening trials, this protocol is suitable for selection in large populations as large amounts of seeds can be screened rigorously in the limited space of the seedling nurseries.

Significant genetic variations in mesocotyl elongation, together with bias distribution in rice germplasm populations, were observed in those screening experiments [6,8,10,11,12,16,17,18,36,37]. Largely different ranges were reported, mainly at the side of high phenotypic values (Appendix A). In addition to the difference caused by screening methods, varied genetic diversity among those populations should be the major reason, implying higher efficiency by using diverse germplasm collections. A higher proportion of germplasm accessions with moderate MLs (with approximately half having an ML > 1 cm) than several other screening results was observed and explained to be caused by different genotypes [16].

### 3.2. Novel QTLs and Candidate Genes Were Identified through GWAS and Linkage Analysis

Mesocotyl length is a quantitative trait and displays a substantial amount of variation among genotypes. Several previous studies have reported the associated QTLs and candidate genes for rice mesocotyl elongation through linkage analysis or GWAS [12,15,16,17,18,20,38]. Some known QTL intervals and novel QTLs could be detected through linkage analysis and GWAS in this study (Appendix A). Among them, few overlapping cases were observed between the associated LD blocks in RDP1 and the CIs of QTLs in the HY73 RIL population. One LD block on rice chromosome 1, i.e., block 605 (marker 5506–5509, physical distance 41,826,155–41,836,778), and two sole SNPs (physical positions 41,956,669 and 41,957,405) were located within the CI of the QTL mapped for ML1, ML2, and ML3 on the end of rice chromosome 1 (physical distance 41,114,455–42,917,379). The LD block 362 (markers 2919–2921, 2923, and 2925–2931; physical distance 27,626,617–27,726,591) was located quite close to the CI of the QTL on chromosome 6 for ML2 and ML3 (Figure 3 and Figure 4, Appendix A). Comparing the candidate genes from our previous GWAS in the mini-core collection of Chinese rice germplasm plus drought-resistant parental lines [12] and the RDP1 population in this study, only one candidate gene (*LOC_Os01g71410*, glycosyl hydrolase family 17) was commonly predicted. This gene did not show differential expression in the light exposure experiment. However, based on the strategy of candidate genes within the LD block, instead of being hit by the top SNP [12], more genes could be predicted as candidates from this associated locus. Thus, more putative candidate genes from the adjacent chromosomal regions, e.g., *LOC_Os01g71340* (glycosyl hydrolase family 17), *LOC_Os01g71350* (glycosyl hydrolases family 17), *LOC_Os01g71420* (Ser/Thr protein phosphatase family protein), *LOC_Os01g71790* (NAM), *LOC_Os01g71820* (glycosyl hydrolase family 17), *LOC_Os01g71830* (glycosyl hydrolase family 17), *LOC_Os01g71860* (glycosyl hydrolase family 17), and *LOC_Os01g71970* (GRAS family transcription factor containing protein), could be included in the candidate gene list to be further investigated as they showed upregulated expression in the mesocotyls of rice seedlings after light exposure, but *LOC_Os01g71420* encoding the Ser/Thr protein phosphatase family protein was downregulated (Appendix A). This chromosomal region also hosted the QTL *qMel-1* that was primarily mapped in the Kasalath/Nipponbare backcross inbred line population and was finely mapped in the derived population from the chromosomal segment substitution line [15]. The importance of this chromosomal region was further supported by more repeatedly mapped QTLs or associated SNPs for mesocotyl length and by being adjacent to the genes of *gy1* and *sd1* [13,19,37].

Within the closely located regions of associated SNPs (LD block 362) and QTLs for ML2 and ML3 on chromosome 6, one candidate gene, *OsFtsH2* (*LOC_Os06g45820*) encoding FtsH protease (homologue of *AtFtsH2/H8*), was upregulated after 60 min light exposure in the mesocotyl of rice seedlings (Appendix A). These QTLs and candidate genes identified in this study will help to better understand the genetic mechanism controlling rice mesocotyl elongation.

### 3.3. Varied Strategies Could Be Considered to Select Donor Parents for DDSR Breeding Programs Concerning More Environmental Issues

The dry direct sowing rice production system faces more unfavorable environmental issues if compared with transplanting rice or wet direct sowing rice cropping. Inadequate rainfall, together with limited irrigation capacity or decreases in irrigation, thus drought proneness, is the inherent scenario of a DDSR production system. Focusing on the seed germination and seedling establishment stage, the soil moisture and temperature are two major environmental factors. A water deficit in the top soil layer (e.g., 0–5 cm) is the most frequent limiting factor delaying seed sowing or causing failed, or partially bad seedling establishment. An earlier study showed an association between mesocotyl elongation and seedling vigor [39]. Rice germplasm collections or segregating populations like RILs were also screened for traits related to seedling vigor [9,40,41,42,43]. As those screening trials were conducted under light and shallow sowing conditions, non-ML-driven seedling vigor traits were detected. Germplasm accessions with mesocotyl elongation, drought resistance, and other seedling vigor traits could be nominated as the donor parents for deep sowing tolerance in DDSR breeding.

## 4. Materials and Methods

### 4.1. Phenotyping Mesocotyl Elongation in the Rice Diversity Panel 1 (RDP1) and HY73 Recombinant Inbred Line (RIL) Population

A subset of 294 accessions with sufficient seeds from 423 accessions in Rice Diversity Panel 1 (RDP1) [23] was phenotypic for mesocotyl elongation. For each accession, 15 brown rice grains were sterilized in 2.5% sodium hypochlorite solution for 20 min, rinsed with sterilized water three times, and then embedded on MS media. The mesocotyl lengths of eight seedlings were measured after germination and growth at 30 °C in darkness for 7 days.

A subset of 312 lines, randomly selected from 1320 lines in the RIL population derived from the commercial hybrid rice Hanyou 73 (HY73, F_1_ of Huhan7A × Hanhui 3), was phenotyped for mesocotyl elongation in three replicated trials (conducted successively). For each line in one trial, 12 brown rice grains were embedded into 12 wells along the long side of the 96-well plate with perforated well bottoms, inserting a black plastic straw in each well to keep the seedling growing straight. The mesocotyl length of seedlings was measured after germination and growth at 30 °C in darkness for 14 days, abbreviated as ML1, ML2, and ML3 for the three trials. All seedlings were exposed to light to prevent subsequent mesocotyl elongation during the period of measurement.

### 4.2. Genotypic Data of RDP1 and HY73 RIL Population

The RDP1 genotypic data of 36,901 SNPs were generated from the Affymetrix SNP array containing 44,100 SNP variants, which is publicly available via the NCBI dbSNP Database (accession codes 469281739 to 469324700) [26].

The whole set of HY73 RIL populations had 1320 lines at the F_9_ generation. Genotyping via the resequencing approach was used to generate a genotypic dataset containing more than 5M SNPs, with the Shuhui 498 genome assembly as a reference (http://www.mbkbase.org/R498/ (accessed on 22 November 2017)) [44]. Following the previously reported pipeline [45], a set of 316,089 high-quality SNPs was used to construct a bin map containing 10,574 bins (Ma XS et al., unpublished data). For the subset of 312 RILs used in this study, a compressed version of the bin map containing 5,571 bins, after merging the adjacent redundant bins, was used for linkage mapping.

### 4.3. GWAS and Linkage Mapping

Using the mean mesocotyl length values of the accessions in the RDP1 subset as the phenotypic data, together with the genotypic data of 36,901 SNPs, a GWAS was conducted using the CMLM algorithm implemented in the software package GAPIT v2.25, using the parameter “PCA.total = 2” [46]. A threshold of FDR-adjusted *p* < 0.05 was used to declare significantly associated markers, including those declared by the stringent threshold of −log_10_(0.05/36901) = 5.868 following Bonferroni correction for multiple tests.

The mean values of mesocotyl lengths of HY73 RILs in each trial (ML1, ML2, and ML3) were used as phenotypic data in linkage mapping. The inclusive composite interval mapping for the additive and dominant QTL (ICIM-ADD) in the software package of QTL IciMapping v4.2 was used to locate the significant QTLs [47]. A threshold of LOD values was determined using permutation tests of 1000 times with *p* < 0.05. The confidence intervals (CIs) were defined using the one-LOD-value drop strategy, where the physical positions on the Shuhui 498 genome were flanked by the first and last bin markers of each CI.

### 4.4. Candidate Gene Prediction

The genotypic data files of RDP1 (sativas413.ped, sativas413.map, and sativas413.fam) were retrieved from the zip file RiceDiversity.44K.MSU6.Genotypes_PLINK.zip downloaded from the website of RiceDiversity.org. Plink was used to transfer the data file to the format compatible for Haploview (e.g., sativas413.chr1.ped, sativas413.chr1.info, and sativas413.chr1.nosex). Haploview was used to estimate the linkage disequilibrium (LD) blocks for each chromosome, using a max LD comparison distance = 100 Kb and maximum amount of missing data allowed per individual = 0.5 [48]. The annotated genes within the LD blocks containing associated SNPs, or hit by sole associated SNPs, were retrieved from the rice genome annotation database v6.1 (http://rice.uga.edu (accessed on 31 May 2021)) to serve as the primary pool of putative candidate genes. The annotated genes within the CIs of QTLs in the HY73 RIL population were also included in this candidate gene pool (Appendix A). As the Nipponbare or Shuhui 498 genome assembly was used as the reference in the genotyping pipelines of the RDP1 and HY73 populations, respectively, the physical positions of the CIs detected in the HY73 population were transferred to the physical positions on the Nipponbare reference genome by blastall 2.2.17 (-p blastn).

Information about overlapping with other gene/QTL mapping results and/or repeated identification in gene transcriptional, functional, or regulating studies was used to predict more promising candidate genes. For instance, our previous transcriptome sequencing experiment measured the differentially expressed genes (DEGs) among the samples of mesocotyl tissue of Zhaxima seedlings in darkness and light exposure conditions for 20 min, 60 min, and 360 min [33]. Zhaxima has an extreme phenotype for ML, so its ML gene expression patterns should be particularly informative about candidate loci. Several publications also reported candidate gene prediction according to their GWAS results for mesocotyl elongation [12,16,36,37]. Wide involvement of multiple plant hormones in regulating mesocotyl elongation, e.g., gibberellin, cytokinin, ethylene, jasmonic acid, strigolactone, and brassinosteroid, was reported in several studies in rice [19,49,50,51,52,53,54]. Similar to the strategy used in previous studies [16,33], those genes from repeatedly detected genomic regions and/or related to phytohormone metabolism or signal transduction would be regarded as candidate genes with higher reliability.

## Figures and Tables

**Figure 1 plants-12-02743-f001:**
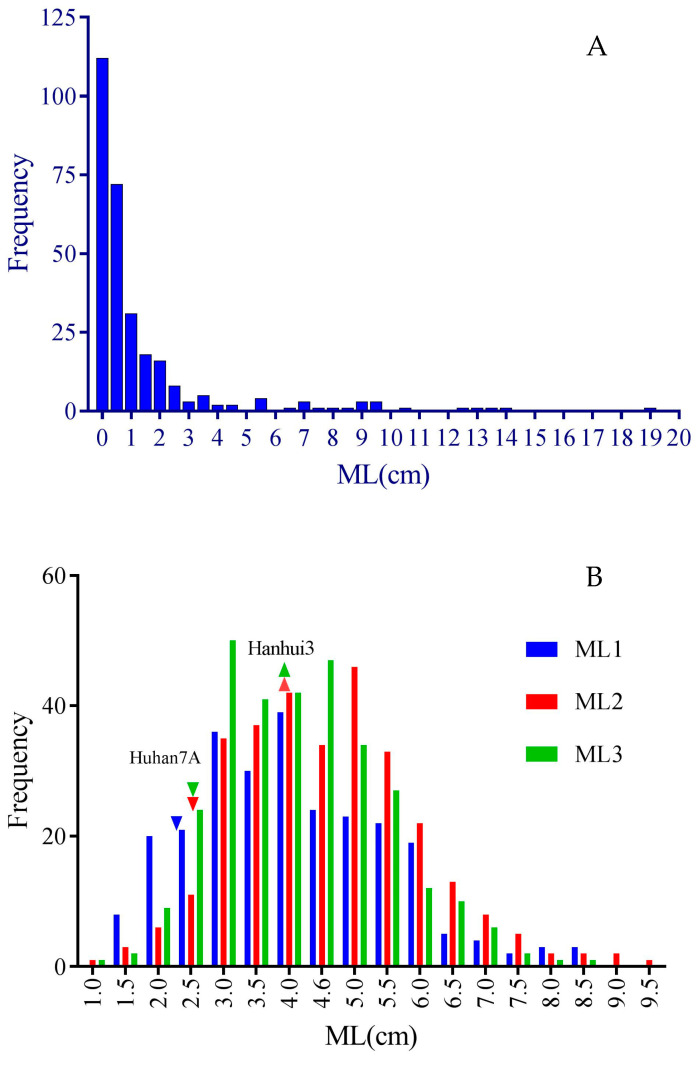
Frequency distribution of mesocotyl lengths measured in subsets of RDP1 (**A**) and HY73 RIL (**B**) populations. Downward and upward triangles represent the average MLs of Huhan 7A and Hanhui 3 (data missed in trial 1), respectively. ML1, ML2, and ML3 represent mesocotyl length measured in three successive trials.

**Figure 2 plants-12-02743-f002:**
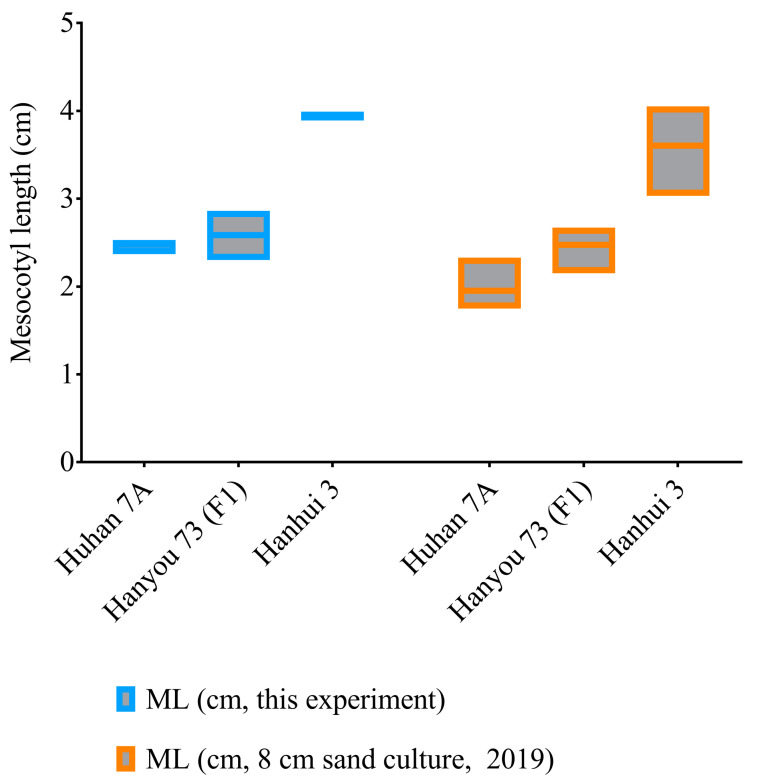
Mesocotyl lengths of the hybrid HY73 and two parental lines, Huhan 7A and Hanhui 3.

**Figure 3 plants-12-02743-f003:**
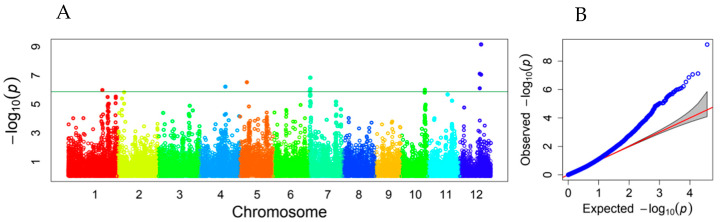
Manhattan plot (**A**) and QQ plot (**B**) showing associated loci for mesocotyl elongation identified in a subset of the RDP1 population. Varied colors represent different chromosomes. The horizontal line represents the threshold of −log10 (0.05/36901) = 5.868.

**Figure 4 plants-12-02743-f004:**
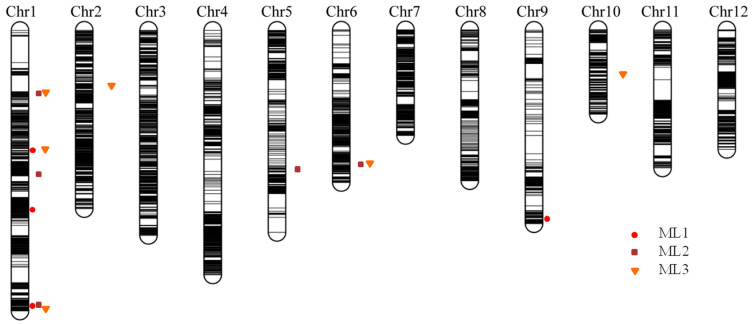
The high-density bin map constructed in a subset of HY73 RIL population and QTLs for elongated mesocotyl lengths measured in three successive trials. ML1, ML2, and ML3 represent the mesocotyl lengths measured in three trials.

**Table 1 plants-12-02743-t001:** ANOVA of mesocotyl elongation measured in subsets of RDP1 and HY73 RIL populations.

Population	Source of Variation	SS (Type III)	*df*	MS	*F* Value	*p* Value
RDP1	Accessions	1,578,722.41	293	5369.80	335.02	0.000
	Residual	26,494.84	1653	16.03		
HY73 RILs	Lines	10,363.36	311	33.32	25.74	0.000
	Trials	442.50	2	221.25	170.92	0.000
	Lines × trials	1913.13	555	3.45	2.66	0.000
	Residual	7750.12	5987	1.29		

**Table 2 plants-12-02743-t002:** QTLs for mesocotyl elongation mapped in HY73 RIL population.

Trait	Chr.	Left Marker	Right Marker	LOD	PVE (%)	Add (cm)
ML1	1	Chr1_bin348	Chr1_bin350	10.68	14.12	0.53
ML1	1	Chr1_bin899	Chr1_bin902	3.58	4.46	0.30
ML1	1	Chr1_bin1342	Chr1_bin1348	3.98	4.92	0.31
ML1	9	Chr9_bin8476	Chr9_bin8481	4.50	5.64	−0.33
ML2	1	Chr1_bin147	Chr1_bin149	7.67	7.69	0.40
ML2	1	Chr1_bin612	Chr1_bin620	12.02	11.96	0.49
ML2	1	Chr1_bin1322	Chr1_bin1327	4.54	4.28	0.29
ML2	5	Chr5_bin5536	Chr5_bin5537	3.24	3.03	0.24
ML2	6	Chr6_bin6431	Chr6_bin6433	7.49	7.24	−0.38
ML3	1	Chr1_bin140	Chr1_bin146	5.78	5.70	0.29
ML3	1	Chr1_bin348	Chr1_bin350	8.51	8.53	0.35
ML3	1	Chr1_bin1396	Chr1_bin1398	5.79	5.74	0.28
ML3	2	Chr2_bin1887	Chr2_bin1895	4.09	3.99	−0.24
ML3	6	Chr6_bin6431	Chr6_bin6433	3.94	3.82	−0.23
ML3	10	Chr10_bin8945	Chr10_bin8950	4.98	4.92	0.29

Chr, chromosome; LOD, logarithm of the odds; PVE, phenotypic variance explained; Add, additive effect. ML1, ML2, and ML3 represent the mesocotyl lengths measured in three trials.

## Data Availability

The data presented in this study are available on request from the corresponding authors.

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
