# Peer review of "Identification of Genetic Loci for Rice Seedling Mesocotyl Elongation in Both Natural and Artificial Segregating Populations"

_plants, 2023, doi:10.3390/plants12142743_

Round 1
Reviewer 1 Report
The manuscript describes the inheritance of mesocotyl length of rice via dark gemination using genome-wide association study, and validated part of the results via RIL population and the dynamic RNA-seq data. The research is well designed and reasonable results were obtained. But there are some aspects to be improved before the publication.
1. The title of the manuscript is not so precise, GWAS is main the content of this study rather than genetic mapping. The author should treat it with caution.
2. The marks A and B were forgot for the figure 1.
3. Line 87, the full name of GWAS, namely the genome-wide association study, has been described previously and do not need to repeat.
4. Format of references should be revised. The name of species should be in italic form.
Lastly, the English writing ofthe manuscript should be extensively revised.
Author Response
Dear Reviewer 1,
Please see the attachment.

Reviewer 2 Report
Identification of genetic loci controlling the mesocotyl elongation of rice seedlings via genetic mapping in both natural and artificial segregating populations
Feng et al, submitted to Plants 2023
The manuscript describes two genetic approaches to identify genetic loci and candidate genes linked to the mesocotyl elongation in rice. The authors have found associations on all 12 rice chromosomes as well as six QTLs. Gene content as well as transcriptomics data from previous studies were used to point out candidate genes.
The manuscript is relatively well organized but the English needs a thorough review. The article is mainly targeted to researchers and breeders from the rice community but could find a wider audience too. However, it suffers from several issues that need to be addressed:
- The results described in the manuscript come from four different experiments led in controlled conditions (one with the RDP1 panel, the three others with the HY73 RILs) according to the material and methods. However, results from a field experiment are also presented and discussed. Please insert the materials and methods for this experiment or take out Figure 5 as well as point 3.1 of the discussion. This point might need complete rewriting, I was just not able to understand the authors reasoning.
- The authors mention a subset of the panel and RIL population were phenotyped. However, we do not know how this subset was selected and whether it is representative of the complete set. Did the sub setting introduce any structure bias, was this taken into account in the analysis ?
- In the methods, the author used a scoring date after 7 days for the panel, but 14 days for the RILs, what was the reasoning behind this difference? Could it make a difference for the genetic components that might be involved?
- In the discussion, the authors point out several co-localization between loci found associated with mesocotyl length in the literature and in their own work. There are a lot of loci involved and it is difficult to point out which are the most important or recurrent ones, if there are any. Moreover, besides co-localization, did the authors check whether the allelic effect identified in different approaches were going the same way?
- For candidate gene prioritization, the authors used transcriptomic data acquired in another study on the genotype Zhaxima. I am not familiar with rice genetic resources and I do not know whether this particular genotype is related to any of the accessions used in the study. Could the author state why these samples are of interest with regards to the genetic studies in this work?
There are many grammatical mistakes all the way through the paper as well as surprising word associations. The paper should be read thoroughly before resubmitting.
Author Response
Dear Reviewer 2,
Please see the attachment.

Reviewer 3 Report
The article reports the genetic loci about mesocotyl elongation by two populations,
Meanwhile, by combining with previous data, the candidate genes were furtherly narrowed down. Undoubtedly, this is a good beginning on the topic of dry direct sowing rice. However, there are still some questions.
Major comments
Q1. Can you provide more evidences for supporting phenotype data about the 6 accessions (>10cm) or 19 accessions (>4cm) in the RDP1. And can you explain how to apply the typical materials in hybrid breeding and the mechanism?
Q2. The phenotype data were not extremely typical (2.5cm vs 4.0cm) in the parents of RIL population. And the F1 (HY73) is 2.5cm in ML. While the phenotype was expanded on 48DAS in figure 5. The content is confusing and misleading.
Q3. Line 383-388.
The package GAPIT includes several models. And the all parameters should be described clearly.
Q4. It can be improved for a more concise title.
Minor Comments
Q5. Line 50. What is the difference between dry direct-seeded rice and dry direct sowing rice?
Q6. In the table1, the number 294 in df was incorrect. And in the line 161, the number 39601 is incorrect.
Q7. The note A and B should be noted clearly in figure 1 and figure 3. And the font-size is inconsistent in figure 2.
Q8. It is very hard to understand the main idea for redundant and meaningless abbreviations in the manuscript, such as WUE and NUE.
The language edition is required.
Author Response
Dear Reviewer 3,
Please see the attachment.
